# Cooperative Effect of Ni-Decorated Monolayer WS_2_, NiO, and AC on Improving the Flame Retardancy and Mechanical Property of Polypropylene Blends

**DOI:** 10.3390/polym15132791

**Published:** 2023-06-23

**Authors:** Mingqiang Shao, Yiran Shi, Jiangtao Liu, Baoxia Xue, Mei Niu

**Affiliations:** College of Textile Engineering, Taiyuan University of Technology, Taiyuan 030024, China

**Keywords:** polypropylene, Ni-modified monolayer WS_2_, sulfur vacancy, activated carbon, NiO, carbon nanotubes, flame retardancy, mechanical properties

## Abstract

Improving the residual char of polypropylene (PP) is difficult due to the preferential complete combustion. Here, we designed a combination catalyst that not only provides physical barrier effects, but also dramatically promotes catalytic charring activity. We successfully synthesized WS_2_ monolayer sheets decorated with isolated Ni atoms that bond covalently to sulfur vacancies on the basal planes via thiourea. Subsequently, PP blends composed of 8 wt.% Ni-decorated WS_2_, NiO, and activated carbon (AC) were obtained (^E^Ni-^S^WS_2_-AC-PP). Combining the physical barrier effects of WS_2_ monolayer sheets with the excellent catalytic carbonization ability of the ^E^Ni-^S^WS_2_-AC combination catalyst, the PP blends showed a remarkable improvement in flame retardancy, with the yield of residual char reaching as high as 41.6 wt.%. According to scanning electron microscopy (SEM) and transmission electron microscopy (TEM) observations, it was revealed that the microstructure of residual char contained a large number of carbon nanotubes. The production of a large amount of residual char not only reduced the release of pyrolytic products, but also formed a thermal shield preventing oxygen and heat transport. Compared to pure PP, the peak heat release rate (pHRR) and total heat release rate (THR) of ^E^Ni-^S^WS_2_-AC-PP were reduced by 46.32% and 26.03%, respectively. Furthermore, benefiting from the highly dispersed WS_2_, the tensile strength and Young’s modulus of ^E^Ni-^S^WS_2_-AC-PP showed similar values to pure PP, without sacrificing the toughness.

## 1. Introduction

Polypropylene (PP) has the advantages of low density, good heat resistance, excellent bending fatigue resistance, chemical stability, and electrical properties. Therefore, it is widely used in automobiles, electronic and electrical appliances, fibers, and other fields [1,2]. Although PP has good heat resistance due to its high thermal decomposition temperature, it is still flammable due to its inherent chemical composition and molecular structure, which limits its wide applications in many fields due to safety considerations [3,4]. Accordingly, the improvement of flame retardancy is a major problem in the application of PP material.

Among many flame retardants, intumescent flame retardants have received extensive research attention due to their excellent flame retardancy [5,6,7,8]. Intumescent flame retardants are mainly composed of three parts, a carbon source, an acid source, and a gas source. They can create a porous and expansive carbon layer during combustion, thereby blocking the transfer of oxygen and the feedback of heat to the surface [9]. However, due to the high addition amount of intumescent flame retardants, which usually reaches 20–40%, such a high level of addition results in a certain impact on the mechanical properties of the material. Therefore, it is significant to develop flame retardants that can form a protective carbon layer to insulate air and heat, without sacrificing the mechanical properties of materials. Currently, carbon nanotubes (CNTs) are widely researched due to their high ratio of length to diameter, as well as excellent flame retardancy [10,11]. For example, introduction of 2% CNT to an epoxy resin system led to a 30.5 % reduction in peak heat release rate (pHRR) [12]. At the same time, it was also found that introducing CNTs coated with ammonium polyphosphate into poly(butylene succinate) not only reduced pHRR by 67%, but also reduced the amounts of CO and CO_2_ by 72.5% and 68%, effectively reducing the emission of harmful gases [13].

Conventionally, CNTs are introduced as additives into polymers via mechanical stirring [12,13]. Presently, it has been proven that CNTs can be produced in situ during combustion, showing highly efficient flame retardancy [14,15]. Unlike the traditional preparation method of CNTs, the in situ formation of CNTs was achieved in which the pyrolytic products from the polymer acted as the carbon source rather than the individual methane, ethylene, or other hydrocarbons [16,17]. For example, Tang et al. reported that the yield of residual char was raised from 0.2% to 18.2% when SiO_2_ and Ni_2_O_3_ were introduced into PP, and the residual char existed in the form of CNTs and carbon nanofibers [15]. In such a system, the “combined catalyst” is mainly composed of two components, including the dehydrogenation and carbonization promoter. Metals such as Pt, Pd, Fe, or nickel components catalyze polymers into dehydrogenation products, and then CNTs are formed under the action of carbonization promoters such as AC, SiO_2_, and montmorillonite [14,15,18,19,20]. Recently, our groups demonstrated that the combination of NiO, Al_2_O_3_, and AC has remarkable charring activity for PP. In PP blends, NiO and AC were used as dehydrogenation and carbonization promoters, respectively. Al_2_O_3_ could promote the dispersion of NiO, as well as enhance its dehydrogenation ability. Unfortunately, similar to other inorganic additives, the introduction of NiO, Al_2_O_3_, and AC into PP also resulted in a decrease in mechanical properties, although the yield of char reached as high as 44.6 wt.%. Therefore, improving the flame retardancy of PP without sacrificing its original excellent mechanical properties, so as to meet the requirements of production and application, has important significance. Noble metals such as Pt and Pd show high hydrogenation/dehydrogenation activity at low reaction temperature; however, they are expensive, hindering their large-scale utilization. Compared to noble metals, the widely used Ni-WS_2_ and Co-MoS_2_ show outstanding hydrogenation/dehydrogenation ability in industrial applications [21,22,23]. In the present study, the Ni-WS_2_ component was selected for investigation, as it typically shows a better hydrogenation/dehydrogenation ability than Co-MoS_2_. However, in bulk WS_2_, the highest proportion of the basal plane is inert, only possessing active sites at the edge, which results in undesirable activity at low reaction temperatures (see Appendix A).

Here, we firstly prepared monolayer WS_2_, which has an abundance of active sites on the basal plane. The Ni-decorated monolayer WS_2_ had high dehydrogenation activity at low temperature, allowing the preferential transformation of pyrolytic products into residual char rather than combustion. The obtained PP blends modified by Ni decorated WS_2_, NiO, and AC showed remarkable flame retardancy efficiency by catalyzing the small molecules degraded from PP into abundant carbon in the form of CNTs. In addition, the single-layer WS_2_ not only promoted the dehydrogenation of long chains into small molecules such as C_2_H_4_ and C_3_H_6_, but also acted as a barrier layer in the form of a carbon nanosheet, hindering heat feedback and oxygen transfer. Furthermore, the introduction of single-layer WS_2_ also enhanced the tensile strength. This study investigated the flame retardancy and mechanical properties of the abovementioned blends. The carbonization mechanism was studied to reveal the synergistic effect of Ni-WS_2_, NiO, and AC on improving the flame retardancy and mechanical properties of PP.

## 2. Materials and Methods

### 2.1. Materials

30 mesh of PP was a commercial product (M_w_ = 29.7 × 10^4^ g/mol, Polydispersity = 3.68, melting point = 166.9 ± 0.5 °C, melt flow index (2.16 kg/190 °C) = 11.2 ± 0.29 g/10 min), density = 0.9054 g/cm^3^, (Huachuang Chemical Co., Ltd. Shanghai, China). 200 mesh of activated carbon, commercial Ni(NO_3_)_2_·6H_2_O, n-butyllithium/hexane (1.6 M), thiourea (analytical reagent, 99%), isopropanol (analytical reagent, 99.7%), bulk WS_2_ (analytical reagent, 99.7%) and polyvinylpyrrolidone (M.W. ≈ 40,000 g/mol) were purchased from Aladdin Biochemical Technology Co. Ltd., (Shanghai, China).

### 2.2. Preparation of Single Layer WS_2_

The prepared method of single layer WS_2_ (^S^WS_2_) is similar to the MoS_2_ [21]. Briefly, 0.3 g of purchased bulk WS_2_ was transferred into a flask loaded with 4 mL of 1.6 M n-butyllithium/hexane under argon atmosphere for 2 days at room temperature, in which the bulk WS_2_ was exfoliated into few layers. The obtained products of LiWS_2_ were diluted with 80 mL of hexane and filtered at reduced pressure to remove the excess butyllithium. After that the obtained powders were poured into 200 mL of water and the resulting suspension was sonicated to assist the exfoliation of few layers of WS_2_ into single layer, which the Li of LiWS_2_ could react with water to produce H_2_ assisting the complete separation of few layers into monolayer. The sonicated suspension was centrifuged at 4000 r.p.m. for 30 min and only the supernatant was collected to remove the unsuccessfully exfoliated bulk WS_2_. Finally, a certain amount of HCl was dropped into the collected supernatant until the PH reaches 7. The precipitate was collected and dried in N_2_ for overnight at room temperature.

### 2.3. Preparation of Ni-Decorated WS_2_

The isolated Ni atom promoted ^S^WS_2_ (^F^Ni-^S^WS_2_) catalyst was prepared using a hydrothermal method, which is similar to the preparation of Co-MoS_2_ [21]. Firstly, 9.33 mg of Ni(NO_3_)_2_·6H_2_O was added into the 100 mL 0.047 mol/L thiourea solution and kept for 12 h to form the cation of Ni(thiourea)_4_^2+^. After that, the 1 mL solution was added into 50 mL 1 mg/L ^S^WS_2_ colloid (30 v/v% isopropanol/water with 50 mg of polyvinylpyrrolidone). The obtained mixed solution was sealed in the 100 mL autoclave and left for 24 h at 160 °C. After the reaction, the precipitate was washed with deionized water and then dried in N_2_ for overnight. In the system, the thiourea acts with sulfur vacancy sites over ^S^WS_2_. To avoid the aggregation of Ni atom, the concentration of Ni complex is low, which the ratio of Ni atom to sulfur vacancy is just 10 times. The prepared procedure was presented in Figure 1.

The material contained Ni decorated ^S^WS_2_ and a certain amount of NiO was namely ^E^Ni-^S^WS_2_. The material contained bulk WS_2_ (^B^WS_2_) and a certain amount of NiO was namely ^E^Ni-^B^WS_2_.

### 2.4. Preparation of PP Blends

A series of PP blends with different composition were synthesized and their detailed composition were listed in Table 1. The different ratio of material was mixed firstly thoroughly in a beaker. The resultant blends were extruded into a sample with a size of 4 × 2 × 0.3 cm^3^ by a vulcanizing press at 180 °C for 15 min (YiTong, Dongguan, China). In the prepared process, the mold was placed in the middle of the vulcanizing press to maintain the flatness of the extruded sample.

### 2.5. Characterization

The phase structures of PP blends before and after combustion were both analyzed by X-ray diffraction (XRD) using TD-3500 diffractometer (Dandong Tongda Technology Co., Ltd., Dandong, China) with Cu Kα radiation at λ = 1.542 Å, giving a voltage of 40 kV and 40 mA. The degree of crystallinity (*X_c_*) of PP is represented by Equation (1):*X_c_* = *I_c_*/(*I_c_* + *I_a_*) × 100(1)
where *I_c_* is the crystalline area and *I_a_* is the amorphous area in the XRD diffractogram.

The Raman spectroscopy was performed on a SPEX-1403 (Renishaw inVia, London, UK) to study the purity and graphitization degree of the produced carbon, and the spectrum was collected at a laser wavelength of 532 nm with Raman shift from 1000 to 2000 cm^−1^. Thermogravimetry (TGA) for PP and its blends were done using a thermal analysis instrument (TG4000, Perkin Elmer, Shanghai, China) from room temperature to 700 °C under air and nitrogen atmosphere with a heating rate of 10 °C/min. The sample measured is 10 mg. An imitative combustion tests were performed to test the residual char. Firstly, a piece of sample (4 × 2 × 0.3 cm^3^) compressed by curing press was placed in a short quartz tube and the tube was pushed to the middle of tubular furnace which was heated at 700 °C. And then the residual char was collected and weighted. Morphologies and microstructure of residual chars of PP blends were observed by means of TEM (FEI Tecnai G2 F20, FEI, Hillsboro, OR, USA and JEM-F200, JEOL, Tokyo, Japan) at 200 kV accelerating voltage. The analyzed samples were placed in ethanol with an ultrasonic dispersion for an hour and deposited on a Cu grid, then the samples were dried overnight. It is noted that the samples after combustion were analyzed without any purification by HF or HNO_3_. The morphologies of the typical samples were recorded by scanning electron microscopy (SEM) (Sigma 500, Zeiss, Oberkochen, German) at 5 KV accelerating voltage. The fire retardant properties of fiber samples were tested with the microcalorimetry (MCC) MCC (MODEL-MCC-3, Deatak, McHenry, IL, USA), at 20 °C/min heating rate under oxygen from 50 to 700 °C using method of SN/T 4879-2017.

## 3. Results and Discussion

### 3.1. Characterization of Monolayer WS_2_

The morphologies of freshly prepared WS_2_ after exfoliation were investigated by TEM microscopy. In Figure 2a, ^B^WS_2_ was successfully exfoliated into a single layer. Approximately 160 flakes from 9 micrographs were analyzed for ^S^WS_2_ sample to obtain relatively accurate statistical results. In the material, 45% of its flakes were present as monolayer, 29% as bi-layers, 16% as tri-layers, and so on. After the incorporation of Ni atoms, the morphology of ^S^WS_2_ still retained its sheet-like structure (Figure 2b).

EPR analysis was used to determine the successful creation of sulfur vacancies over the WS_2_ surface, by measuring the unpaired electrons on coordinatively unsaturated defective sites. The ^B^WS_2_ showed abundant number of sulfur vacancies (0.98 × 10^17^ g^−1^), which were derived from the edges and crystallite interfaces (see in Appendix A). As compared to ^B^WS_2_, the number of defective sites in case of ^S^WS_2_ as detected by EPR reached 3.86 × 10^17^ g^−1^, indicating the successful creation of large number of sulfur vacancies.

### 3.2. Dispersion States of Ni and WS_2_ in PP Matrix

Wide-angle XRD was used to investigate the dispersion and composition of PP blends and the results are presented in Figure 3. The diffraction peak at 22° in AC was ascribed to the amorphous carbon. The peaks at 14.2°, 17.0°, 18.6°, and 21.8° in the PP blends were the characteristic of PP [24]. Compared to ^B^WS_2_, the intensity of ^S^WS_2_ was dramatically reduced and its full width at half maxima value increased, indicating the successful exfoliation of ^B^WS_2_ into few- or single-layers. The diffraction peaks at 14.4°, 33.5°, and 58.8° corresponded to the (002), (101), and (110) planes of WS_2_ (PDF#35-0651), respectively. After the addition of ^B^WS_2_, all the ^B^WS_2_-PP, ^B^WS_2_-AC-PP, and ^E^Ni-^B^WS_2_-AC-PP samples showed the characteristic peaks of WS_2_. In contrast, after addition of ^S^WS_2_, the samples showed no obvious diffraction peaks of WS_2_, indicating the high dispersivity of WS_2_. The dispersion of additives in the matrix is considered to be one of most important factors that influence the physical and chemical properties, including mechanical properties and flame retardancy. Generally, 2D nanoadditives like graphene and transition-metal dichalcogenides (e.g., MoS_2_, WS_2_) exhibit better flame retardancy than 0D nanoadditives (e.g., carbon dot) and 1D (e.g., carbon nanowires), due to their superior physical barrier property. Unfortunately, the poor interfacial compatibility between the 2D material and polymer results in unsatisfactory flame retardancy performance. The introduction of thiourea in the system, not only helped the interaction of thiourea with PP matrix, but also acted with its abundant number of sulfur vacancies at the edge and over and the surface of WS_2_. Finally, the high dispersion of WS_2_ was achieved in the cases of ^F^Ni-^S^WS_2_-AC-PP and ^E^Ni-^S^WS_2_-AC-PP, due to the high compatibility between the filler of WS_2_ and PP matrix. In ^E^Ni-^S^WS_2_-AC-PP and ^E^Ni-^B^WS_2_-AC-PP samples, the weak diffraction peaks at 48.1° and 56.1° were ascribed to the (111) and (200) planes of Ni (PDF#04–0835), indicating that part of Ni atom suffered due to self-nucleation during the hydrothermal process. At the same time, the characteristic peaks at 46.8° and 68.3° ascribed to NiO, were also observed for both ^E^Ni-^S^WS_2_-AC-PP and ^E^Ni-^B^WS_2_-AC-PP samples, which was the result of oxidation of Ni to NiO.

The degree of crystallinity influences the thermal stability and mechanical property of materials. The crystallinities of PP and PP blends were calculated by XRD and the results are presented in Table 1. Compared to neat PP, AC-PP had higher degree of crystallinity, since AC acted as a nucleating agent and promoted the crystal growth of PP [25,26]. In contrast, the introduction of ^B^WS_2_ not only served as the crystal nucleus but also hindered the growth of crystal region of PP. In ^E^Ni-^B^WS_2_-AC-PP sample, the crystallinity decreased from 42.1 to 39.5%. Similar to ^B^WS_2_, some particles of NiO introduced were larger than 30–40 nm, which could not act as nucleating agents any more [27]. In contrast to the influence of ^B^WS_2_, the addition of 2D monolayer of WS_2_ into PP system in case of ^F^Ni-^S^WS_2_-AC-PP and ^E^Ni-^S^WS_2_-AC-PP could dramatically improve the degree of crystallinity, since the induced monolayer of WS_2_ acted as a nucleating agent [28,29].

### 3.3. Thermal Decomposition Behaviors

It is important to investigate the combustion behaviors of materials in terms of thermal decomposition temperature. TGA is commonly used to evaluate the thermal stabilities of the polymers. The thermal stabilities of PP and its blends in air and N_2_ atmosphere were determined by TGA, and the results are presented in Figure 4 and Figure 5. The values of onset decomposition temperature (*T*_5%wt_), 10 wt% decomposition temperature (*T*_10%wt_), 50 wt% decomposition temperature (*T*_50%wt_), and maximum decomposition temperature (*T*_max_) are listed in Table 2 and Table 3. In air atmosphere, the *T*_5%wt_, *T*_10%wt_, *T*_50%wt_, and *T*_max_ values of pure PP were 265.9, 285.8, 344.1, and 356.5 °C, respectively. After introduction of ^B^WS_2_, there was no obvious enhancement in the decomposition temperature. However, the PP modified with AC showed increase in *T*_5%wt_, *T*_10%wt_, *T*_50%wt_, and *T*_max_ values by 40.6, 41.3, 29.7, and 18 °C, respectively. The introduction of AC into the PP could not only trap the radicals but also promote the crosslinking reactions and curing degree of PP [12]. In the case of both ^F^Ni-^S^WS_2_-AC-PP and ^E^Ni-^S^WS_2_-AC-PP, the decomposition temperature was dramatically improved, which could be attributed to the physical barrier effect of monolayered WS_2_ through hinderance and transfer of heat and decomposition products. Furthermore, an important difference was noted between ^F^Ni-^S^WS_2_-AC-PP and ^E^Ni-^S^WS_2_-AC-PP. In terms of *T*_5wt%_ and *T*_10wt%_, ^F^Ni-^S^WS_2_-AC-PP and ^E^Ni-^S^WS_2_-AC-PP showed similar decomposition temperature, with temperature differences of only 1.7 and 4.4 °C, respectively. However, this difference increased to 12.4 and 12.6 °C for the *T*_50wt%_ and *T*_max_ values. In the previous study, it was proved that the combination of Ni/NiO and AC could promote the production of residual char during combustion [12]. Compared to ^F^Ni-^S^WS_2_-AC-PP, more of residual char was produced over ^E^Ni-^S^WS_2_-AC-PP, which was an increase from 9.93 to 19.29%. The residual char also acted as a physical barrier that insulated the transmission of heat, which resulted in a remarkable improvement at high decomposition temperature.

The residual masses obtained in air and N_2_ atmosphere are listed in Table 2 and Table 3, respectively. Compared to neat PP, PP modified with AC or WS_2_ alone also showed little residual char. In spite of the combination of WS_2_ and AC, the ^B^WS_2_-AC-PP still showed low yield of residual char, indicating that there was no synergistic effect between ^B^WS_2_ and AC. In addition, it was also observed that introduction of small number of Ni species into ^S^WS_2_-AC-PP, did not dramatically improve the residual char yields. In case of ^E^Ni-^S^WS_2_-AC-PP, the residue yield was somewhat higher than that of ^F^Ni-^S^WS_2_-AC-PP, which was the result of improvement in carbonization ability with the interactions between Ni/NiO and AC. Noteworthily, the introduction of abundant number of NiO species into ^S^WS_2_-AC-PP led to significant improvement in the residual char yields to as high as 21.29 wt%, demonstrating the synergistic effect that improved the residual char yield and thermal stability. The trends for decomposition temperature and residue measured under nitrogen atmosphere were similar.

### 3.4. Characterization of Residual Char

SEM was used to obtain the detailed information about the microstructures of PP blends after imitative combustion at 700 °C and the results are presented in Figure 6(a1–a5). As shown in Figure 6(a1–a3), the residue was made up of amorphous carbon. Based on the yields of imitative combustion, the amorphous carbon was mainly derived from the added AC rather than produced from the combustion of PP. Compared to ^F^Ni-^S^WS_2_-AC-PP, the morphology of ^E^Ni-^S^WS_2_-AC-PP after combustion was totally different. Interestingly, a large number of CNTs were observed. Moreover, no amorphous carbon was seen, since it could have been covered with CNTs. Overall, the dimensions of CNTs were not uniform and part of the CNT was crooked and intertwined, which was consistent with the previous report [20]. In the case of ^E^Ni-^B^WS_2_-AC-PP, only small amount of filamentous carbon (marked by red dashed cycles) was found. Interestingly, further investigation of the external surface of CNTs (Figure 6I) showed that the surface was coarse with many protrusions. In contrast to the CNTs produced from specific single components like C_2_H_4_ or C_3_H_6_, the pyrolytic products were complex and combustion was rapid and violent, which resulted in the asymmetrical growth of CNTs. The results of TEM analysis demonstrated that the filamentous carbon was of CNTs too. Based on the vibrational modes of Raman spectroscopy, the graphitization degree of the combustible products was analyzed. Among them, the *D*-band at 1350 cm^−1^ represented the impurities in the sidewall structure and the *G*-band was associated with the order of graphene layers of carbon nanotubes [30]. Therefore, the higher the ratio of *I*_G_ to *I*_D_, the higher was the graphitization degree. From the results of Raman spectroscopy in Figure 7, *I*_G_/*I*_D_ was lower than 1, suggesting that the graphitization degree of CNTs was not high. Comparatively, when the monolayered WS_2_ was replaced by ^B^WS_2_, only little proportion of filamentous carbon was observed.

TEM analysis was performed to obtain detailed information regarding the microstructures of the residues of PP blends after combustion experiment and the results are presented in Figure 6(b1–b5). The residues of ^B^WS_2_-PP and ^B^WS_2_-AC-PP were amorphous, mainly comprising of either WS_2_ or AC. In the case of ^F^Ni-^S^WS_2_-AC-PP, WS_2_ was observed after combustion except for the amorphous carbon. As shown in the Figure 6(b3), even though the aggregation phenomenon was obvious after combustion at high temperature, large amounts of single- and few- layered structures were still observed together. The high dispersivity acted as an efficient physical barrier and showed catalytic activity. In the case of ^E^Ni-^S^WS_2_-AC-PP, a large amount of product in the form of CNTs was observed (See Figure 6(b4)). In the image, the CNTs had a wide range of lengths (0.1–1.1 µm). The diameter of CNTs varied between 4 nm and 25 nm. Interestingly, it is observed that some particles had sizes smaller than the diameter of CNTs (marked by white dashed- circles) and the particle remained preferably encapsulated in CNTs. At the same time, some particles had sizes larger than the diameter of CNTs and the CNTs grew along with the particle brim (marked by orange dashed-circles). Furthermore, the relationship between CNTs and Ni particle was evident in Figure 6, wherein the smaller particles were encapsulated, whereas larger particles were located at the edge of CNTs.

### 3.5. Mechanical Properties

Figure 8 shows the representative tensile strain-stress curves of PP and its composites and the detailed data is presented in Table 4. It was evident that after the addition of AC or ^B^WS_2_ alone, both the tensile strength and Young’s modulus decreased, as compared with neat PP. Furthermore, the extent of decrease in case of ^B^WS_2_-PP was slightly higher than that of AC-PP. Compared to ^B^WS_2_, AC had greater number of groups and better interactions with PP matrix, which caused lesser damage to the mechanical property of PP. When both, ^B^WS_2_ and AC, were introduced, the tensile strength and Young’s modulus did not show significant enhancement, indicating that the absence of dramatic synergistic effects between ^B^WS_2_ and AC on the mechanical property. The tensile strength and Young’s modulus of ^F^Ni-^S^WS_2_-AC-PP were dramatically increased, which was higher than that of pure PP. The 2D materials (carbon nanosheet, MoS_2_, etc.) are known to be excellent fillers that improve the mechanical property [25,26,27,28,29,30]. WS_2_ has lamellar structure similar to that of MoS_2_, and it also acts as an excellent reinforcement material [29,31]. Furthermore, during synthesis, WS_2_ with abundant number of sulfur vacancies was introduced into the PP matrix with the assistance of thiourea. The ^S^WS_2_ coordinated with thiourea had good compatibility with PP and retained high dispersivity. Therefore, the tensile strength and Young’s modulus showed remarkable improvement. In the case of ^E^Ni-^S^WS_2_-AC-PP, 4 wt% of NiO was introduced into the PP. The network structure was reported to be formed due to the interactions between the metallic oxides and AC, which was beneficial for the improvements in flame retardancy and mechanical property [14]. Thus, ^E^Ni-^S^WS_2_-AC-PP still showed higher values of tensile strength and Young’s modulus than pure PP, although abundant number of NiO were introduced. The tensile strength and Young’s modulus were also slightly lower than those of ^F^Ni-^S^WS_2_-AC-PP, suggesting that NiO had a negative influence on the mechanical property. Compared to the changes in tensile strength, the difference in elongation at break values for different PP blends was more obvious. After addition of ^B^WS_2_, the elongation at break dramatically decreased to 18%, a reduction of 87.7%, as compared with pure PP. This was the result of poor interfacial interaction between large-sized ^B^WS_2_ particles and PP matrix. A similar phenomenon was seen when AC was added to the PP matrix, wherein the elongation at break was reduced by 78.1%, as compared with pure PP. After introduction of both AC and ^B^WS_2_, a remarkable improvement was observed, which could be attributed to the better dispersion of ^B^WS_2_. ^F^Ni-^S^WS_2_-AC-PP and ^E^Ni-^S^WS_2_-AC-PP showed significant differences in the elongation at break values. The elongation at break for ^E^Ni-^S^WS_2_-AC-PP increased by 10.9%, as compared with neat PP. This was ascribed to the network structure formed from NiO and AC. The network structure not only promoted the stretching in the vertical direction, but also improved the elongation in the horizontal direction [14].

### 3.6. Flame Retardancy and Flame Retardant Mechanism

MCC test, based on the working principle of oxygen consumption, is one of the most effective methods to investigate the combustion properties of polymer materials. It provides the values of heat release capacity (HRC), peak heat release rate (pHRR), total heat release (THR), and temperature at the peak heat release rate (*T*_max_) [32,33,34,35]. The recorded parameters are very important as they reflect the combustion properties of materials. HRR curves of neat PP, ^B^WS_2_-PP, ^B^WS_2_-AC-PP, ^E^Ni-^B^WS_2_-AC-PP, ^F^Ni-^S^WS_2_-AC-PP, and ^E^Ni-^S^WS_2_-AC-PP are presented in Figure 9 and the detailed data is presented in Table 5. Only one peak was observed for each system, which was similar to that of DTG results. From Figure 9, it could be inferred that pure PP produced more combustible products during its heating with a sharp peak for HRR and pHRR of 1317 w/g, which suggested rapid combustion and high spread rate of fire. Compared to pure PP, pHRR values from the plots of ^B^WS_2_-PP and ^B^WS_2_-AC-PP blends were reduced by 6.07% and 14.12%, respectively, as compared to that of neat PP. This indicated that the flammability of PP was to some extent reduced on addition of WS_2_ or AC. After the introduction of abundant number of NiO into ^B^WS_2_-AC-PP, the HRR was further reduced by 23.16%. In the case of ^F^Ni-^S^WS_2_-AC-PP, the pHRR (1084 W/g) was somewhat higher than that of ^E^Ni-^B^WS_2_-AC-PP. However, the *T*_pHRR_ of ^E^Ni -^B^WS_2_-AC-PP was lower than that of ^F^Ni-^S^WS_2_-AC-PP. This indicated that HRR decreased after the addition of NiO into the system. Additionally, the individual use of single-layered WS_2_ and AC without the assistance of rich NiO could not prevent the generation of combustible gases. After the introduction of a large amount of NiO into the ^S^WS_2_-AC-PP, the pHRR value was much lower than that of the other system, demonstrating the best synergistic effect.

From Table 5, the *T*_pHRR_ values for PP blends modified with different components were, to some extent, higher than that of pure PP. Especially both, ^F^Ni-^S^WS_2_-AC-PP and ^E^Ni -^S^WS_2_-AC-PP, showed dramatic enhancement, implying that the introduction of single-layered WS_2_ formed a physical barrier that blocked the transfer of heat. The *T*_pHRR_ values showed trends similar to those of *T*_max_ values, as obtained by TGA. Additionally, in Figure 9, the ignition temperatures of neat polymer and its blends showed little differences. The order of onset temperature was ^B^WS_2_-AC-PP, ^E^Ni-^B^WS_2_-AC-PP, ^B^WS_2_-PP, PP, ^F^Ni-^S^WS_2_-AC-PP, and ^E^Ni-^S^WS_2_-AC-PP. Among them, the onset temperature of ^B^WS_2_-PP was almost the same as that of pure PP, indicating that the presence of ^B^WS_2_ had no significant influence on the onset of the combustion process. The onset temperatures of ^B^WS_2_-AC-PP and ^E^Ni-^B^WS_2_-AC-PP were lower than those of ^B^WS_2_-PP and PP. This was due to the fact that AC promoted the cracking of PP fragment radicals into smaller molecules [36]. In addition, the ^F^Ni-^S^WS_2_-AC-PP and ^E^Ni-^S^WS_2_-AC-PP benefited from the good physical barrier effect that prevented the transfer of heat, resulting in a higher onset temperature.

Table 5 presents the THR values of PP and its blends. In contrast to PP, the THR values of ^B^WS_2_-PP and ^B^WS_2_-AC-PP blends were 40.2 and 39.5 kJ/g, respectively, which corresponded to reduction by 7.37% and 8.99%, respectively. The THR values of ^F^Ni-^S^WS_2_-AC-PP and ^E^Ni-^B^WS_2_-AC-PP were further reduced by 10.36% and 12.90%, respectively, implying that some portions of PP the blends were protected without complete combustion. After the introduction of abundant number of NiO into ^S^WS_2_-AC-PP, the THR values showed remarkable influence and were reduced to 32.1 kJ/g, corresponding to a decrease by 26.03%. This suggested that the effective combination of multiple components led to a significant improvement in flame retardancy of the blends. The HRC was also an important parameter, which was useful to predict and evaluate the combustion behavior [33,34]. The results are presented in Table 5. Pure PP had the highest HRC value of 1241 J/g K, while ^B^WS_2_-PP and ^B^WS_2_-AC-PP showed lower HRC values. The addition of Ni into the PP blends led to further reduction of HRC value. The HRC values of ^F^Ni-^S^WS_2_-AC-PP, ^E^Ni-^S^WS_2_-AC-PP, and ^E^Ni-^B^WS_2_-AC-PP were 1002, 643, and 996 J/g K, respectively, which showed similar changing trends as that of pHRR values.

Based on the results of imitative combustion experiments (Table 1) and MCC test, it was evident that the yield of residual char and flame retardancy were dramatically improved for ^E^Ni-^S^WS_2_-AC-PP, as compared with ^F^Ni-^S^WS_2_-AC-PP and ^E^Ni-^B^WS_2_-AC-PP. This suggested that ^E^Ni-^S^WS_2_-AC combination catalyst possessed good synergistic effects. Hence, the absence of rich NiO or replacement of monolayered WS_2_ with ^B^WS_2_ led to unsatisfactory performance. In this system, the flame-retardancy action of ^E^Ni-^S^WS_2_-AC in the PP matrix was mainly achieved due to the synergistic effects of the physical barrier property of WS_2_ and the excellent ability of catalytic carbonization (Figure 10). On one hand, WS_2_ was introduced into the PP matrix through the assistance of thiourea, which prevented the aggregation during the subsequent molding process using vulcanizing press. During the burning of PP blends, the highly dispersed WS_2_ acted as 2D material and formed an efficient physical barrier that prevented the transfer of heat and mass. On the other hand, the high amounts of decomposition products of PP were carbonized to CNTs due to the synergistic effect of ^E^Ni-^S^WS_2_-AC, thus decreasing the amount of combustible material. The formed CNTs also acted as protective char layers that prevented the feedback of heat and transfer of oxygen and pyrolytic products.

Some researchers have put forth the plausible mechanism for the growth of CNTs from PP by combination of nickel and montmorillonite or AC [14,15,20,37]. It was concluded that the process of catalytic carbonization of PP into CNTs could be achieved in three steps due to the synergistic action of ^E^Ni-^S^WS_2_-AC combination catalyst [38,39,40,41,42,43,44]. First, at low temperature of about 200–300 °C, the surface of PP blends gradually degraded into smaller molecules due to the action of Ni modified WS_2_ and acid sites. At the same time, NiO was reduced to metallic Ni by hydrogen, methane, etc. Subsequently, the smaller molecules like alkynes, olefins, and other hydrocarbons were adsorbed on the surface of Ni particles and then were decomposed to carbon atoms. In addition, the hydrogen produced further brought about the reduction of NiO to Ni. Finally, due to the synergistic action of AC, the carbon atoms migrated along with the Ni particles and interconnected with each other to produce dimmers and trimmers. Then, the dimmers and trimmers further reacted to form five-to six-membered rings and graphite sheets. The graphite sheets continuously grew into CNTs along with the PP pyrolytic products. Due to the complex nature of the decomposition products and conditions, the growth rate of graphite sheets was not regular, resulting in the low degree of graphitization.

Generally, ^E^Ni-^B^WS_2_ shows high hydro/dehydrogenation activity at high temperatures of about 360 °C [23]. However, at low temperature (about 200–300 °C), the hydro/dehydrogenation ability was poor. Furthermore, the active sites of ^E^Ni-^B^WS were mainly located at the edge, while the basal plane was inert (see in Appendix A). The Ni-modified monolayer WS_2_ not only had abundant number of active sites over the plane, but also showed excellent hydro/dehydrogenation performance at low temperature. This allowed the preferential transformation of pyrolytic products into CNTs before combustion. In summary, in the case of ^E^Ni-^S^WS_2_-AC-PP, the high activity of dehydrogenation at low temperature over the Ni-modified single layer WS_2_ could provide as many olefins and alkynes to the reduced Ni particle as possible. Further, the olefins and alkynes were further decomposed and grew over the reduced Ni particle. With the assistance of AC, the carbon atoms grew into CNTs alongside the Ni surface. Therefore, the TEM image showed that the Ni particle was encapsulated in the CNTs or located at both the ends or tightly attached to the carbon nanotubes. Compared with ^E^Ni-^S^WS_2_-AC-PP, ^F^Ni-^S^WS_2_-AC-PP lacked sufficient number of Ni particles, which failed to efficiently transform the small molecules into carbon atoms. Meanwhile, in the case of ^E^Ni-^B^WS_2_-AC-PP, the ^E^Ni-^B^WS_2_ had low dehydrogenation ability and could not enrich with olefins and alkynes, though it had abundant number of Ni species.

## 4. Conclusions

In this work, Ni-decorated WS_2_ monolayer sheets were obtained using Li ion intercalation and hydrothermal method. XRD and TEM results showed that the WS_2_ was well-dispersed in the matrix. The ^E^Ni-^S^WS_2_-AC combination catalyst showed excellent flame retardancy through barrier effects and catalytic charring activity. In ^E^Ni-^S^WS_2_-AC-PP system, WS_2_ layer acted as a physical barrier and the abundant amount of residual char was produced due to the synergistic effects of ^E^Ni-^S^WS_2_-AC. From SEM and TEM analyses, the residual char was mainly found to be composed of CNTs. CNTs were considered to be in situ produced during combustion by dehydrogenation of pyrolytic products over Ni-decorated monolayer WS_2_. Then, the carbon atoms grew into CNTs along with the surface of Ni particle through the assistance of AC. The protective barrier composed of CNTs could remarkably reduce the release of combustible products and suppress the heat feedback and oxygen transfer between the surface and matrix. Compared to pure PP, the peak heat release rate (pHRR), total heat release rate (THR) of ^E^Ni-^S^WS_2_-AC-PP were reduced by 46.3% and 26.0%, respectively. In addition, the tensile strength and Young’s modulus were slightly higher than those of pure PP. This strategy not only promoted the flame retardancy of PP, but its effectiveness can also be extended to other polymer systems and even valuable carbon or other nanomaterials can be obtained from biodegradable polymers.

## Figures and Tables

**Figure 1 polymers-15-02791-f001:**
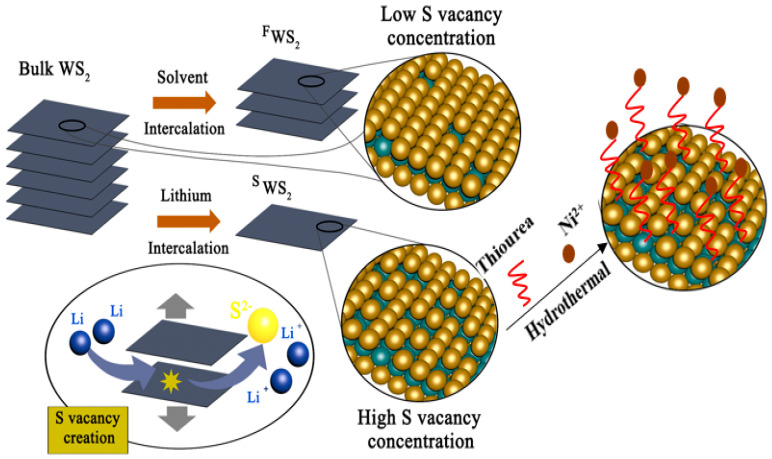
Preparation routes of Ni-decorated WS_2_ monolayer sheets by physical (via solvent intercalation), chemical (via Li intercalation) and hydrothermal methods.

**Figure 2 polymers-15-02791-f002:**
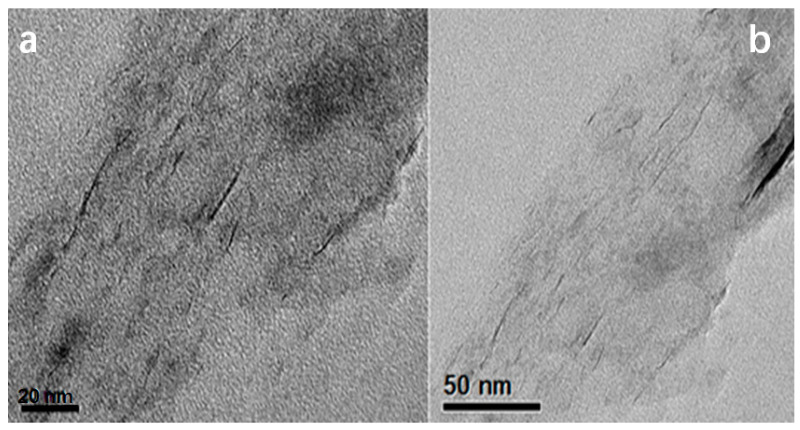
TEM image analysis of fresh sample before (**a**) and after (**b**) the decoration of Ni.

**Figure 3 polymers-15-02791-f003:**
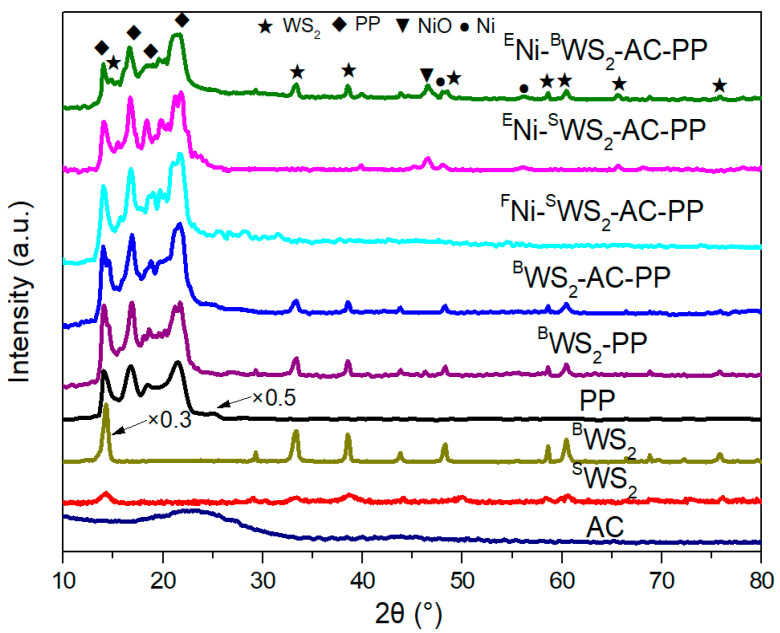
XRD patterns of neat AC, ^S^WS_2_, ^B^WS_2_, neat PP and PP blends.

**Figure 4 polymers-15-02791-f004:**
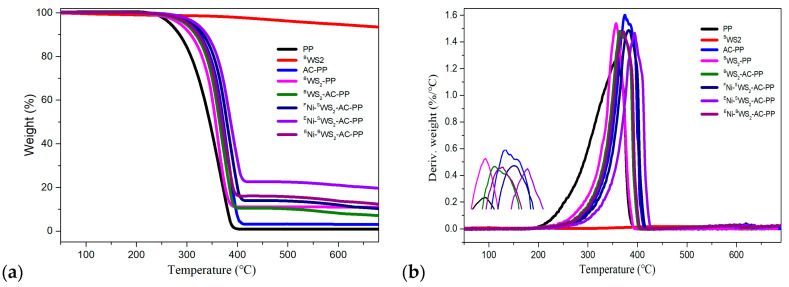
(**a**) TGA and (**b**) DTG curves of PP and its blends at 10 °C/min in air atmosphere.

**Figure 5 polymers-15-02791-f005:**
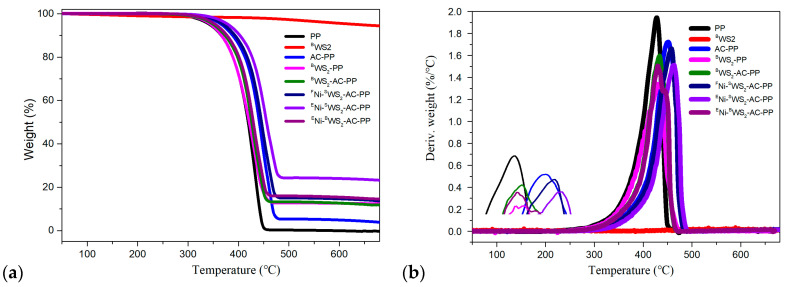
(**a**) TGA and (**b**) DTG curves of PP and its blends at 10 °C/min in N_2_ atmosphere.

**Figure 6 polymers-15-02791-f006:**
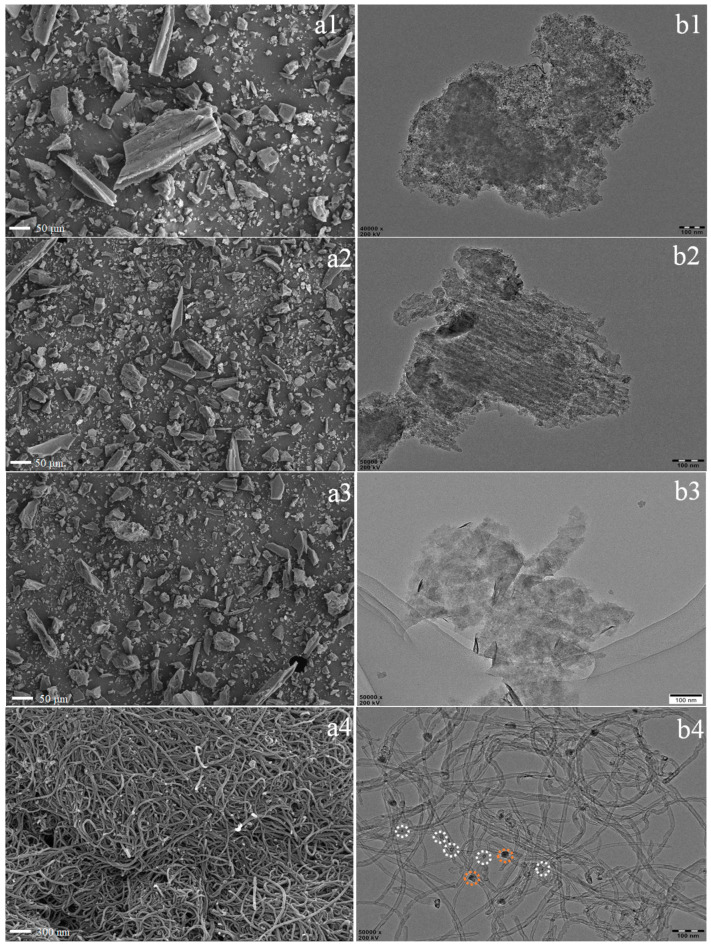
SEM images of char residues for PP composites after imitative combustion: (**a1**) AC-PP; (**a2**) ^B^WS_2_-AC-PP; (**a3**) ^F^Ni-^S^WS_2_-AC-PP; (**a4**) ^E^Ni-^S^WS_2_-AC-PP; (**a5**) ^E^Ni-^B^WS_2_-AC-PP. TEM images of char residues for PP composites after imitative combustion: (**b1**) AC-PP; (**b2**) ^B^WS_2_-AC-PP; (**b3**) ^F^Ni-^S^WS_2_-AC-PP; (**b4**) ^E^Ni-^S^WS_2_-AC-PP; (**b5**) ^E^Ni-^B^WS_2_-AC-PP. I and II are different magnification of SEM and TEM images of char residues for ^E^Ni-^S^WS_2_-AC-PP.

**Figure 7 polymers-15-02791-f007:**
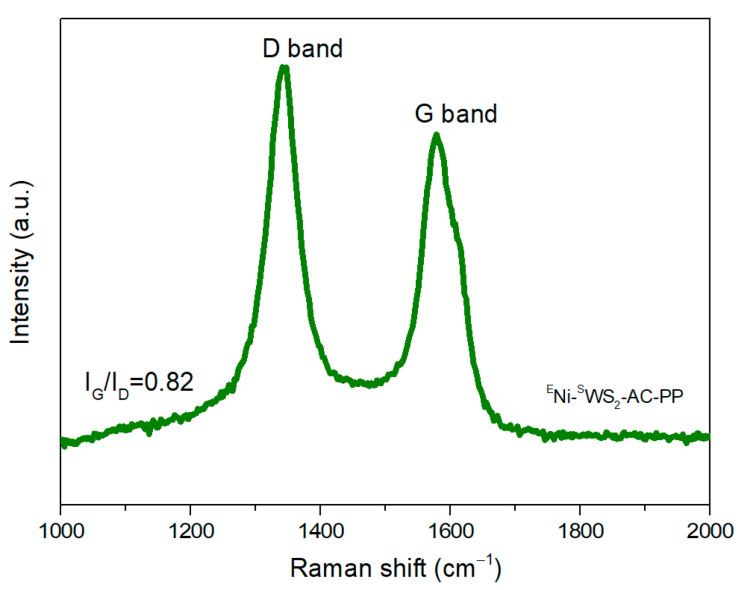
Raman spectrum of ^E^Ni-^S^WS_2_-AC-PP blends after imitative combustion.

**Figure 8 polymers-15-02791-f008:**
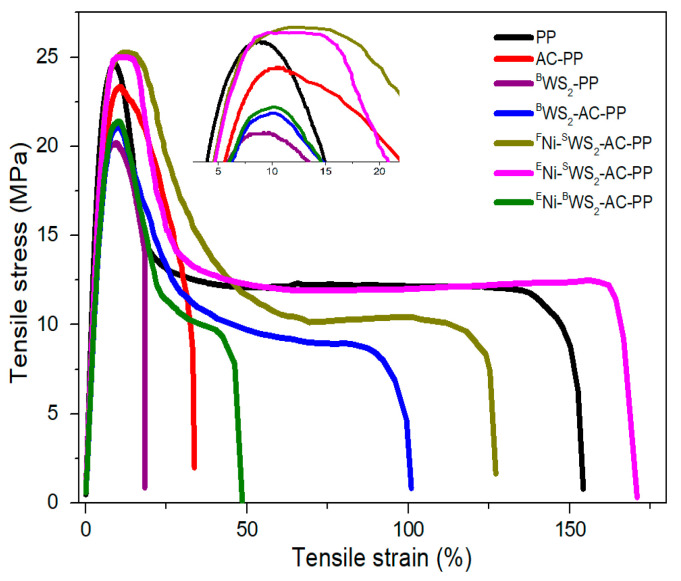
Tensile stress–strain curves of PP and its blends.

**Figure 9 polymers-15-02791-f009:**
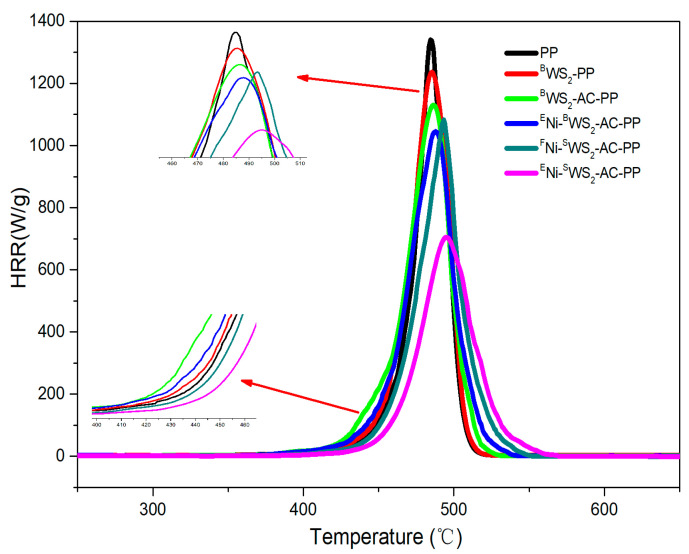
HRR curves of PP and PP blends. Insets are enlarged initial low temperature decomposition between 400 and 460 °C and the peak of HRR.

**Figure 10 polymers-15-02791-f010:**
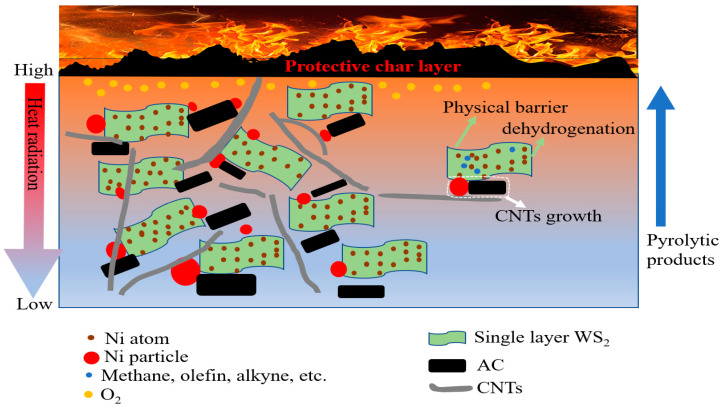
A proposed flame-retardant mode-of-action of the synergistic effect of ^E^Ni-^S^WS_2_-AC on improving the flame retardancy of PP.

**Table 1 polymers-15-02791-t001:** Formulations of PP blends, degree of crystallinity, and the yield of residual chars from imitative combustion.

Sample	PP (g)	AC (g)	WS2 (g)	Ni (Coordinated with Thiourea) (mg)	NiO (g)	Sum (g)	Residue (wt%)
PP	100	-	-	-	-	100	0
BWS2	-	-	100	-	-	100	-
AC-PP	92	8	-	-	-	100	3.1
BWS2-PP	92	-	8	-	-	100	8.6
BWS2-AC-PP	92	4	4	-	-	100	6.4
FNi-SWS2-AC-PP	92	4	4	8.4 × 10^−2^	-	≈100	8.3
ENi-SWS2-AC-PP	92	2	2	8.4 × 10^−2^	4	≈100	41.6
ENi-BWS2-AC-PP	92	2	2	0	4	100	14.9

**Table 2 polymers-15-02791-t002:** Summary of TG and DTG data of PP and its blends in air atmosphere.

Sample	T_5%wt_ ^a^	T_10%wt_ ^b^	T_50%wt_ ^c^	T_max_ ^d^	Residue (wt%)
PP	265.9	285.8	344.1	356.5	0.33
^B^WS_2_	577.9	-	-	-	93.36
AC-PP	306.5	327.1	373.8	374.5	3.05
^B^WS_2_-PP	268.3	293.9	345.4	356.9	9.94
^B^WS_2_-AC-PP	298.3	318.8	366.5	367.8	8.57
^F^Ni -^S^WS_2_-AC-PP	312.4	332.5	381.7	383.1	9.93
^E^Ni-^S^WS_2_-AC-PP	314.1	336.9	394.1	395.7	21.29
^E^Ni-^B^WS_2_-AC-PP	302.4	322.9	369.7	373.1	12.73

^a^ Temperature at 5 wt% mass loss; ^b^ Temperature at 10 wt% mass loss; ^c^ Temperature at 50 wt% mass loss; ^d^ Temperature at maximum mass loss rate.

**Table 3 polymers-15-02791-t003:** Summary of TG and DTG data of PP and its blends in N_2_ atmosphere.

Sample	T_5%wt_	T_10%wt_	T_50%wt_	T_max_	Residue (wt%)
PP	342.9	365.9	418.8	427.6	0.21
^B^WS_2_	638.3	-	-		94.26
AC-PP	367.6	389.5	443.1	451.3	3.91
^B^WS_2_-PP	345.6	366.3	421.1	429.3	12.45
^F^Ni-^S^WS_2_-AC-PP	370.3	393.6	445.9	458.2	10.76
^E^Ni-^S^WS_2_-AC-PP	374.6	397.9	455.6	464.7	23.15
^E^Ni-^B^WS_2_-AC-PP	353.8	375.9	429.3	439.8	14.69

**Table 4 polymers-15-02791-t004:** Summary of the mechanical test results of PP and its blends.

Sample	Tensile Strength	Elongation at Break (%)	Young’s Modulus (MPa)
PP	24.6 ± 0.6	146 ± 15	601 ± 82
AC-PP	23.4 ± 0.4	32 ± 4	568 ± 53
^B^WS_2_-PP	20.1 ± 0.5	18 ± 2	487 ± 61
^B^WS_2_-AC-PP	21.1 ± 0.7	91 ± 11	533 ± 49
^F^Ni-^S^WS_2_-AC-PP	25.4 ± 0.3	120 ± 14	624 ± 84
^E^Ni-^S^WS_2_-AC-PP	25.1 ± 0.5	162 ± 5	609 ± 75
^E^Ni-^B^WS_2_-AC-PP	21.4 ± 0.6	43 ± 4	546 ± 68

**Table 5 polymers-15-02791-t005:** Summary of MCC data for PP and PP blends.

Sample	pHRR (W/g)	pHRR Reduction (%)	T_pHRR_ (°C)	HRC (J/g K)	THR (kJ/g)
PP	1317	-	483.5	1241	43.4
^B^WS_2_-PP	1237	6.07	485.6	1150	40.2
^B^WS_2_-AC-PP	1131	14.12	487.1	1086	39.5
^F^Ni-^S^WS_2_-AC-PP	1084	17.69	493.2	1002	38.9
^E^Ni-^S^WS_2_-AC-PP	707	46.32	495.6	643	32.1
^E^Ni-^B^WS_2_-AC-PP	1012	23.16	488.3	996	37.8

## Data Availability

The data presented in this study are available on request from the corresponding author.

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
