# Peer review of "Cooperative Effect of Ni-Decorated Monolayer WS2, NiO, and AC on Improving the Flame Retardancy and Mechanical Property of Polypropylene Blends"

_polymers, 2023, doi:10.3390/polym15132791_

Round 1

Reviewer 1 Report

Manuscript Number: polymers-2391987

Title: Synergistic effect of Ni-decorated monolayer WS2, NiO and AC on improving the flame retardancy and mechanical property of polypropylene composites

The article deals with synthesis of WS2 monolayer sheets decorated with isolated Ni atoms that is subsequently used to prepare polypropylene composites that contains Ni-decorated WS2, NiO and activated carbon. The flame retardant and mechanical properties of the prepared composite were systematically investigated in addition to the carbonization ability and mechanism of flame re. The article is well structured, and the results presented therein are systematically discussed. This research will be useful and will advance the field of polymer composite and polymer materials. Thus, I recommend accepting the article for publication, but after the following minor revision.

The authors should consider the following.

·         In 2.4: Please provide information about the preparation procedure of the PP composites. The formulation of the PP composites are provided in Table 1 but the procedure for the preparation is not provided.

·         Figure 3: The materials in the figure caption should be rearrange in accordance with the XRD patterns in the figure. AC, BWS2, SWS2, PP and PP composites

·         Page 4, line 142: the spectrometer should be replaced with more appropriate word “spectroscopy”.

·         Page 4, line 144: the word “spectra” should be changed to “spectrum”. Spectrum is used for single and the plural is spectra.

·         Page 5, Line 195: Letter B of WS2 should be supper script to be consistent with the other ones.

·         Figure 7 caption: replace the word “spectras” with more appropriate word “spectrum”

Best regards

Author Response

Thank you very much for comments about our manuscript again. The comments are valuable and helpful for revising and improving our manuscript. We have browsed the comments carefully and made revision. We enclosed a revised manuscript according to the comments. The revised portion was marked in red in the manuscript. Furthermore, the PP composites were prepared easily without special means. The image below was the vulcanizing press used in the present work.

Best wishes

Reviewer 2 Report

The manuscript by these Authors deals with the investigation as regards the flammability and mechanical properties of polypropylene composites.
Below please find my review:

1. In line 81, "Figure S1" is incorrect.
2. How was the MFR determined in line 98? At what temperature and under what load? Why is it different from MFI on line 98?
3. In "2.2.1 Preparation of single layer WS2" there is no information about the temperature/temperatures at which the preparation was carried out. As a result, the reaction cannot be reproduced.
4. In the section "2.4.1 Preparation of PP composites" the conditions for obtaining composites are poorly described and in little detail. There is no indication of the equipment used along with the process conditions, individual heating zones, the rotational speed of the screws, etc. It cannot be reproduced.
5. Which figure shows the EPR results? Why is Figure 3 in Section 2.4.1? It should be under the text from point 3.2.
6. The following principle should be followed: description of the discussion and results and the figure or table to which it refers below the description. In the current form, the discussion and the form of presenting the results are mixed up.
7. How do you know sulfur vacancies are 0.98 × 1017 g-1? This is not shown in any Figure.
8. How does the interplanar distance between the filler layers in the composite change?
9. Why was XRD analysis not performed in the range of e.g. 3-10 2Theta?
10. No significant tensile strength result was obtained. However, the effect of the additive on the stretching of the matrix is visible, as susceptibility to deformation, but this is missing in the description.
11. The MCC flammability tests did not present the total burn time of the sample, flash point, and flame extinction. After all, according to the presented Figure 9, it can be seen that the cost of lowering the flammability is the extension of the burning time. The question is how much is the burn time change and how does the flash point of the sample shift relative to the reference sample?

Author Response

  Thank you very much for comments about our manuscript entitled “Synergistic effect of Ni-decorated monolayer WS2, NiO and AC on improving the flame retardancy and mechanical property of polypropylene composites” submitted to Polymers.

  The comments are valuable and helpful for revising and improving our manuscript. We have browsed the comments carefully and made revision. We enclosed a revised manuscript according to the comments. The revised portion was marked in red in the manuscript. Furthermore, some issues about English expression was revised and marked in purple in the manuscript.

1. In line 81, "Figure S1" is incorrect.

  Thank you for your suggestion. It is rewritten in the manuscript.

2. How was the MFR determined in line 98? At what temperature and under what load? Why is it different from MFI on line 98?

  Thank you for your careful guidance. The MFI and MFR are same. The value of MFR was incomplete and the value has been deleted.

3. In "2.2.1 Preparation of single layer WS2" there is no information about the temperature/temperatures at which the preparation was carried out. As a result, the reaction cannot be reproduced.

  Thank you for your suggestion. The detailed information about temperature has been added into the experiment part of section 2.2 in the revised manuscript.

4. In the section "2.4.1 Preparation of PP composites" the conditions for obtaining composites are poorly described and in little detail. There is no indication of the equipment used along with the process conditions, individual heating zones, the rotational speed of the screws, etc. It cannot be reproduced.

  Thank you for your suggestion. The PP composites were prepared by vulcanizing press, which is heated at 180 ℃ for 15 min. The equipment image was presented in Figure 1, which do not have the rotation function. We also added the detailed information about the equipment and process condition in the revised manuscript.  

5. Which figure shows the EPR results? Why is Figure 3 in Section 2.4.1? It should be under the text from point 3.2.

  Thank you for your suggestion. The figure about EPR was presented in supplementary information. And the format was rearranged as you suggested below.

6. The following principle should be followed: description of the discussion and results and the figure or table to which it refers below the description. In the current form, the discussion and the form of presenting the results are mixed up.

  Thank you for your suggestion. The format was rearranged as you suggested.

7. How do you know sulfur vacancies are 0.98 × 1017 g-1? This is not shown in any Figure.

  Thank you for your suggestion. The figure was presented in supplementary information. The calculation was conducted according to the resonance from the amount of microwave radiation absorbed.

8. How does the interplanar distance between the filler layers in the composite change?

  Thank you for your suggestion. The change of interplanar distance between the filler layers was different for the different components. In the present study, we do not know the specific change. Thank you for your suggestion again. We will continue to strive for the research on this suggestion in the future.

9. Why was XRD analysis not performed in the range of e.g. 3-10 2Theta?

  Thank you for your suggestion. According to the PDF cards of XRD, the main 2θ diffraction peaks of various component including NiO, WS2, PP and AC is ranged from 10-80. Therefore, the information in the range of 10-80° is adequate to analyse the material. In addition, we also give some similar reference about XRD analysis.

[1] J. Gong et al. Polymer Degradation and Stability, 99 (2014) 18-26

[2] X.-L. Pu et al. Polymer Degradation and Stability, 206 (2022) 110170

10. No significant tensile strength result was obtained. However, the effect of the additive on the stretching of the matrix is visible, as susceptibility to deformation, but this is missing in the description.

  Thank you for your suggestion. It is rewritten about the elongation at break in the section 3.5.

11. The MCC flammability tests did not present the total burn time of the sample, flash point, and flame extinction. After all, according to the presented Figure 9, it can be seen that the cost of lowering the flammability is the extension of the burning time. The question is how much is the burn time change and how does the flash point of the sample shift relative to the reference sample?

  Thank you for your suggestion. During the imitative combustion using the tube furnace, we had tried to confirm the flash point and burning time. However, it is difficult to confirm the accurate time due to that smoke could be produced before ignition. Furthermore, the tube furnace is long, resulting in difficult distinguish for the flash point and burning time.

  Thank you for your suggestion again. Presently, we are trying to produce the monolayer WS2 with simpler method to obtain the WS2 in large scale compared with the present means. This will allow us to test the flame retardancy with cone calorimeter and obtain the information about the flash point and burning time. And you will see the relative research in the future.     

Thanks for your very valuable and helpful comments again.

Reviewer 3 Report

This work deals with the improvement of residual char of polypropylene (PP)  The authors  designated a combination catalyst which not only provide the  physical barrier effects, but also dramatically promote the catalytic charring activity. They also synthesized WS2 monolayer sheets decorated with isolated Ni atoms that bond covalently to sulfur vacancies on the basal planes by thiourea. The authors used advanced experimental techniques and the whole work is interesting.

POINTS FOR IMPROVEMENT:

1. Please, report specific applications for this composite.

2. A major problem is the chemical and shock stability of the composite. Please, make an appropriate comment.

3. An other point is the health limitations of this composite as in the preparation heavy metals are used. These limitations could be reported.

4. This composite could be recycled after use? Please, make a comment

5. Please, report the crystallinity of PP both as pure reagent and after treatment to produce the composite.

6. The product density of the PP as reagent could be reported.

7. Please, complete the literature or/and the keywords, as a literature review made by the reviewer by using GOOGLESCHOLAR and the provided keywords revealed 4,800 similar works.

Author Response

  Thank you very much for comments about our manuscript entitled “Synergistic effect of Ni-decorated monolayer WS2, NiO and AC on improving the flame retardancy and mechanical property of polypropylene composites” submitted to Polymers.

  The comments are valuable and helpful for revising and improving our manuscript. We have browsed the comments carefully and made. We enclosed a revised manuscript according to the comments. The revised portion was marked in green in the manuscript.

1. Please, report specific applications for this composite.

  Thank you for your suggestion. Polypropylene (PP) has been widely used in many fields such as electronic and electric industry, automobiles and housing due to its good mechanical properties, excellent electrical resistance, low toxicity, low density and good chemical resistance. Presently, the PP-modified by AC and inorganic MoS2 and NiO. The material retains the original mechanical property and improve the flame retardancy. Furthermore, the additive of MoS2, NiO and AC do not have toxicity. Therefore, the PP-modified composites could also be used in the field of electronic and electric industry, automobiles and housing. For example, in the case of automotive interior, it could be used as used for seat and roof materials. For the housing application, it could be used as wall cloth. However, we must acknowledge the drawback of the prepared PP composites that the color is black rather than transparent, restricting the part application.  

2. A major problem is the chemical and shock stability of the composite. Please, make an appropriate comment.

  Thank you for your suggestion. For the flame retardant, the additive of organic small molecules may migrate and cause damage on the flame retardancy and environment. However, in the present work, the additive is composed of AC and inorganic MoS2 and NiO. The additive has large scale compared with organic small molecules. Usually, our materials are used at room temperature. At room temperature, the additive maintain stability and do not aggregate in the PP matrix, resulting in the high chemical and shock stability.

  In summary, this is a good suggestion. Sometimes, the PP material is used at high temperature, which may lead to the aggregation of monolayer WS2 and then influence the flame retardancy. Thanks for the suggestion. We will pay attention to the observation of stability in the future work, especially at high temperature.

3. An other point is the health limitations of this composite as in the preparation heavy metals are used. These limitations could be reported.

  Thank you for your suggestion. As mentioned above, the PP-modified composite has good chemical and shock stability due to the difficult of migration of MoS2 and NiO. Therefore, it will not generate serious hazards. At the same time, the PP-modified composite is usually covered by other material in the application of automobile and housing, which not direct contact with human body.    

4. This composite could be recycled after use? Please, make a comment

  Thank you for your suggestion. After combustion, the residual char is mainly composed of CNTs, NiO, WS2 and AC. According to the TEM results, the NiO particle was encapsulated in CNTs after combustion. Therefore, it is difficult to recover the NiO. However, the suggestion of recycled use is a good idea. Although the AC act as charring promoter after use suffer from loss, the residual char is mainly composed of CNTs or other carbon, which could also act as carbonization promoter. Furthermore, CNTs themselves also have good flame retardancy. From TEM, WS2 after use still retain good monolayer and few layers. Therefore, the composite after use may also have good flame retardancy. We will conduct the corresponding experiment to test the performance in the future. Thanks for your valuable suggestion again.  

5. Please, report the crystallinity of PP both as pure reagent and after treatment to produce the composite.

  Thank you for your suggestion. The crystallinity of PP both as pure reagent and after treatment was calculated according to the XRD results. And the description was presented in the revised manuscript.

6. The product density of the PP as reagent could be reported.

  Thank you for your suggestion. The density of the PP as reagent was added into the section 2.1.

7. Please, complete the literature or/and the keywords, as a literature review made by the reviewer by using GOOGLESCHOLAR and the provided keywords revealed 4,800 similar works.

  Thank you for your suggestion. The key words were modified and rewritten in the manuscript.

Reviewer 4 Report

Polymers 2391987

In this paper the authors provide a combination catalyst for the improvement of residual polypropylene (PP). Synthesized WS2 monolayer in presence of Ni atoms, obtained NI-WS2-AC-PP. The behaviour is analysed by SEM and TEM techniques. The composites show remarkable improvement of the flame retardancy, and the yield of residual char.

Tis paper in interesting. The introduction and references are good but is necessary that the authors reviewed the presentation of the manuscript. A minor revision is necessary.

Comment

1)      Revise of presentation of the manuscript

2)      Line 58, change Tang et. al for Tang et al.

3)      Line 123, change complex for cation

4)      The phase structures were analysed by XRD or PXRD?

5)      Line 141-142, Line 155-156, is the same phrase. Delete one

6)      Part 3.2, Revise the nomenclature. It’s confusing. Revise all manuscript

7)      Revise the presentation of the Figure 4 and 5

8)      Line 272, delete “As shown in Figure 6 a1-b1”

9)      Line 286, define IG and ID

10)   Line 308, change 0.1-1.1um for 0.1-1.1 um

11)   Line 353, define the letters MCC

12)   Revise the presentation of Figure 9

13)   Line 441, change 360ºC[23].. for 360 ºC [23].

14) Revise the English

Revise the English

Author Response

  Thank you very much for comments about our manuscript entitled “Synergistic effect of Ni-decorated monolayer WS2, NiO and AC on improving the flame retardancy and mechanical property of polypropylene composites” submitted to Polymers.       

  The comments are valuable and helpful for revising and improving our manuscript. We have browsed the comments carefully and made revision. We enclosed a revised manuscript according to the comments. The revised portion was marked in orange in the manuscript.

  We are very sorry for our incorrect writing. According to the comments from you, we polished the manuscript with a professional assistance in writing, conscientiously. And the corrected sentences were rewriteen and marked in purple in the manuscript.

  Furthermore, we want to answer the question that the phase structures were analysed by XRD or PXRD?

  In the present work, the characterization was performed by XRD. We also found some references about the similar XRD analysis [1, 2].

[1] J. Gong et al. Polymer Degradation and Stability, 99 (2014) 18-26

[2] X.-L. Pu et al. Polymer Degradation and Stability, 206 (2022) 110170

Round 2

Reviewer 2 Report

it could be accept in present form.

Author Response

Thank you for your affirmation. Thank you

Reviewer 3 Report

Nice WORK.

Author Response

(The authors gave the same response as above.)
